# Changes in SOD and NF-κB Levels in *Substantia Nigra* and the Intestine through Oxidative Stress Effects in a Wistar Rat Model of Ozone Pollution

**DOI:** 10.3390/antiox13050536

**Published:** 2024-04-27

**Authors:** Selva Rivas-Arancibia, Erika Rodríguez-Martínez, Marlen Valdés-Fuentes, Alfredo Miranda-Martínez, Eduardo Hernández-Orozco, Citlali Reséndiz-Ramos

**Affiliations:** Departamento de Fisiología, Facultad de Medicina, Universidad Nacional Autónoma de México, Mexico City 04510, Mexico; ana_ery@yahoo.com.mx (E.R.-M.); marlen_valdez@yahoo.com.mx (M.V.-F.); yobainmx@yahoo.com (A.M.-M.); eduardo_iq@outlook.com (E.H.-O.); tai.citla_03@hotmail.com (C.R.-R.)

**Keywords:** O_3_, oxidative stress, *substantia nigra*, intestine, 4HNE, NFκB, superoxide dismutase

## Abstract

This work aimed to elucidate how O_3_ pollution causes a loss of regulation in the immune response in both the brain and the intestine. In this work, we studied the effect of exposing rats to low doses of O_3_ based on the association between the antioxidant response of superoxide dismutase (SOD) levels and the nuclear factor kappa light chains of activated B cells (NFκB) as markers of inflammation. Method: Seventy-two Wistar rats were used, divided into six groups that received the following treatments: Control and 7, 15, 30, 60, and 90 days of O_3_. After treatment, tissues were extracted and processed using Western blotting, biochemical, and immunohistochemical techniques. The results indicated an increase in 4-hydroxynonenal (4HNE) and Cu/Zn-SOD and a decrease in Mn-SOD, and SOD activity in the substantia nigra, jejunum, and colon decreased. Furthermore, the translocation of NFκB to the nucleus increased in the different organs studied. In conclusion, repeated exposure to O_3_ alters the regulation of the antioxidant and inflammatory response in the substantia nigra and the intestine. This indicates that these factors are critical in the loss of regulation in the inflammatory response; they respond to ozone pollution, which can occur in chronic degenerative diseases.

## 1. Introduction

Environmental pollution caused by O_3_ is a public health problem in highly polluted regions [1]. O_3_, when repeatedly inhaled at low doses, as in the case of environmental pollution, raises the formation of reactive oxygen species (ROS) in the body [2,3,4]. Enzymatic antioxidant systems, such as SOD, catalase, glutathione system, thioredoxin, etc. [5], cannot counteract ROS produced at low doses. Therefore, this causes a state of chronic oxidative stress, which is a key factor in the evolution of degenerative diseases like cancer, diabetes, autoimmune diseases, neurodegenerative diseases, etc. [2,5,6]. Currently, a large number of works describe the pathophysiological mechanisms of environmental pollution in the central nervous system and other organs and systems. In addition, there is a wealth of evidence on the effect of environmental contamination from O_3_ on the brain and, in particular, on the *substantia nigra* [4,7]. These works show that O_3_ contamination, by inducing oxidative stress, plays a fundamental role in the oxidation of dopamine and the alteration of antioxidant systems, leading to the formation of dopamine adducts and triggering the death of dopaminergic neurons [4]. This degenerative process is accompanied by alterations in the intestine and an inflammatory response [8] that loses its signal, making the progressive neurodegeneration process irreversible [2].

On the other hand, the architecture of the intestinal epithelium is very complex; it is formed by a monolayer of cells connected by different cell junctions. This barrier fulfills multiple functions, including the absorption of nutrients and some electrolytes, and provides an immune response. Therefore, it is considered a highly selective semi-permeable barrier used to prevent the free passage of certain toxins, antigens, and microbial products [2,9]. In addition, the loss of homeostasis in intestinal barrier components enables the free passage of substances that can be harmful to the organism, promoting the activation of immune response mechanisms both cellular and molecular [10,11,12,13]. However, studies show that the loss of regulation in the inflammatory response is a critical factor in the loss of intestinal permeability and the generation of chronic pathologies, including neurodegenerative diseases [2,8,14]. Among the main proinflammatory responses is the NFκB transcription factor, which is essential for regulating intestinal epithelial homeostasis and its pathophysiology [15,16].

NFκB is a transcription factor that regulates the expression of genes associated with immune and inflammatory responses, survival, antioxidant systems, and other cellular processes. Oxidative stress activates NFκB synthesis through the following several mechanisms: (1) ROS, such as hydrogen peroxide (H_2_O_2_), can directly activate an inhibitor of kappa kinase (IKK) B, which phosphorylates the inhibitory protein IκB; (2) the NFκB factor is activated by oxidative stress, genotoxic stress, and DNA damage through several different pathways. One of the known routes is through the SUMOylation of the nuclear factor (NF)-κB essential modulator (NEMO, IKK-γ), which is translocated to the nucleus and forms the ataxia-telangiectasia mutated (ATM) kinase ATM/NEMO; when this complex translocates to the cytoplasm, ATM phosphorylates IKKs, releasing NFκB, enabling NFκB translocation to the nucleus and initiating cytokine synthesis [17,18,19,20]. Some works associate the participation of NFκB in intestinal and brain structure disorders related to damage to the *substantia nigra*, especially in Parkinson’s disease.

A complex functional relationship between the brain and the intestine has been widely demonstrated, which is established through the *vagus* nerve, as well as neurotransmitters, hormones, and cytokines that modify the functions of both systems [8]. The brain–gut axis uses chemical messengers like adrenaline, acetylcholine, neurohormones, cortisol, dopamine, and peptides that regulate eating behavior, as well as serotonin and peptides that participate in digestion [21,22]. The loss of regulation in the gut–brain axis is caused by both intrinsic factors (antibiotics, stress, unhealthy diets, etc.) and extrinsic factors (environmental pollution) that produce changes in the microbiota directly affecting the intestinal wall, which contributes to the loss of regulation in the inflammatory response and an increase in the permeability of the intestinal wall, changing the signaling that reaches the brain through different pathways [20,23,24].

The role of chronic oxidative stress produced by pollution in the loss of intestinal permeability and the process of progressive neurodegeneration in brain structures is still unclear. The objective of this work was to elucidate how O_3_ contamination causes a loss of regulation in the immune response both in the intestine and in the *substantia nigra*. For this purpose, the effect of oxidative stress caused by exposure to low doses of O_3_ on the association between the antioxidant response of SOD and the translocation of NFκB to the nucleus is explored as a marker of inflammation in the mentioned structures.

## 2. Materials and Methods

### 2.1. Animals

Seventy-two male rats of the Wistar strain (obtained from the vivarium of the Facultad de Medicina, UNAM) weighing 250 to 300 g were used and were kept individually in acrylic boxes, fed with croquettes for rodents (LabDiet 5001, CA-USA) ad libitum, under constant temperature and humidity conditions, and with a 12:12 light–dark photoperiod. The care of the animals used in this study strictly complied with the Guide for the Care and Use of Laboratory Animals National Research Council [25] and the Official Mexican Standard, NOM-062 [26], on the “Technical specifications for the production, care, and use of laboratory animals”, adjusted to the international guidelines of ethics and animal management aimed at minimizing the number of animals used and their suffering. The project was submitted to the research committee and research ethics committee of the Facultad de Medicina, UNAM, based on Comité Interno para el Cuidado y Uso de Animales de Laboratorio. División de Investigación. Facultad de Medicina 2018 (CICUAL) Registry 019-2023 [27]; the project was approved with registration number FM/DI/063/2023.

### 2.2. Exposure to Ozone

The exposure method for low doses of O_3_ is described in Pereyra-Muñoz N et al., 2006 [4], and Rivas-Arancibia et al., 2010 [28]. Briefly, the rats were placed in a transparent acrylic chamber, and through an O_3_ generator, the independent groups were exposed to 0.25 ppm of O_3_ for 4 h daily for 7, 15, 30, 60, and 90 days. The control group was exposed to 4 h of ozone-free air daily for 30 days. O_3_ levels were kept constant and controlled throughout the experiment with an O_3_ monitor (PCI Systems of O_3_ and Control, West Caldwell, NJ, USA). After inducing deep anesthesia with 50 mg/Kg of sodium pentobarbital, the animals were euthanized per NOM-033 [29]. The brains and intestines were removed and processed for immunohistochemistry, Western blot, and superoxide dismutase activity.

### 2.3. Immunohistochemistry

The brains and intestines were extracted and were fixed in 4% paraformaldehyde. Subsequently, the tissues were dehydrated and embedded in paraffin blocks, and 5 µm thick sections were made and mounted on slides. The tissues were deparaffinized and then hydrated, and an antigen retriever was used (Biocare Medical, Concord, CA, USA). Peroxidase activity was inhibited with 3% H_2_O_2_, and blocking was performed to reduce the background (Background Sniper, 4plusDetection, Biocare Medical, CA, USA). Antibodies against SOD1 (GTX100554, Irvine, CA, USA), SOD2 (GTX116093, Irvine, CA, USA), and NFκB (GTX102090, Irvine, CA, USA) were incubated overnight at 4 °C. Slides were treated with biotinylated secondary antibody (Universal Link, Biocare Medical). They were then incubated with streptavidin (4plus Detection Component, Streptavidin-HRP, Biocare Medical) incubated with 3,3-Diaminobenzidine Substrate chromogen (DAB Kit, ScyTek, West Logan, UT, USA) and stained with hematoxylin. Each slide was analyzed with a BX41 Olympus Microscope and photographed with an Evolution-QImagin Digital Camera Kit (MediaCybernetics, Silver Spring, MD, USA).

### 2.4. Western Blot

Lipid oxidation and abundance of Cu/Zn SOD and Mn SOD in the *substantia nigra*, jejunum, and colon were evaluated:

The *substantia nigra*, jejunum, and colon samples were homogenized in lysis buffer with protease inhibitors (Complete, Roche^®^ Basel, Switzerland), and protein quantification was performed using the BCA method (Micro BCA Protein Assay Kit^®^, Thermo Scientific, Waltham, MA, USA). The proteins were then separated by electrophoresis on a 10% SDS acrylamide gel under reducing conditions. Blotting was performed using a PVDF membrane and blocked using 5% skim milk in TBST. The primary antibody Anti-4HNE (MA5-27570 of Invitrogen^®^, Carlsbad, CA, USA), SOD1 (GTX100554, Irvine, CA, USA), SOD2 (GTX116093, Irvine, CA, USA), and Beta-Actin (GTX110564, Irvine, CA, USA) by GENETEX^®^ were incubated overnight at 4 °C at a concentration of 1:1000. After three washes with TBST, the membranes were incubated with the appropriate secondary antibodies, mouse anti-rabbit or goat anti-mouse, diluted 1:10,000 (Santa Cruz Biotechnology, Dallas, TX, USA). Membranes were developed using 1 mL of Immobilon^®^ Forte Western HRP Substrate Reagent (Millipore^®^ Corporation, Bedford, MA, USA) for 1 min and digitized using the GelCapture^®^ program (v 7.0.5. DNR Bio-Imaging System, Modi’in-Maccabim-Re’ut, Israel). Densitometric analysis was performed using the Image Studio ^®^ software (v 5.2.5. LI-COR Bioscience, Lincoln, NE, USA). All bands were quantified based on their molecular weight, except for 4HNE, which, due to its characteristics, used the entire band for analysis.

### 2.5. Superoxide Dismutase Activity

Measurement of total SOD activity involves inhibition of nitroblue tetrazolium (NTB) reduction by xanthine oxidase, used as a superoxide generator. It is calculated as U/µg of tissue (SOD unit is defined as the amount of enzyme that inhibits the reduction in NTB by 50%) [30]. The *substantia nigra*, jejunum, and colon tissue fragments were homogenized in a protein extraction buffer to which protease inhibitors (Roche^®^, Basel, Switzerland) were added and centrifuged at 15,000 rpm for 20 min at 4 °C. The supernatants were recovered, and the quantification of total proteins was carried out using the Micro BCA Kit (Thermo Scientific, Waltham, MA, USA). The reaction medium was prepared with 0.3 mM xanthine solution (Sigma, St. Louis, MO, USA), 0.6 mM EDTA (J.T. Baker, Philadelphia, PA, USA), 150 µM nitroblue tetrazolium solution (Sigma, St. Louis, MO, USA), 400 mM Na_2_CO_3_ solution, and 0.1% bovine serum albumin (MP Biochemicals, Irvine, CA, USA). Xanthine oxidase (Sigma, St. Louis, MO, USA) was prepared in 2 M (NH_4_)_2_SO_4_. The tissue and the reaction medium were mixed in a tube; xanthine oxidase was added at 15 s intervals between each tube. Each sample was incubated for 15 min at 27 °C, and the reaction was stopped by adding 0.8 mM CuCl_2_. Blank solutions containing deionized water were used instead of tissue. The absorbance of each sample was determined at 560 nm on an Epoch spectrophotometer (Biotek, Winooski, VT, USA).

### 2.6. Statistical Analysis

A Kolmogorov–Smirnov normality test was used to analyze all results. Results are presented as median and interquartile ranges for non-parametric variables. As the data did not provide a normal distribution, the Kruskal–Wallis test was used to see differences between groups, followed by the Mann–Whitney U test to compare the control group with the groups that received different treatments. Differences between groups were considered statistically significant, with a *p* ≤ 0.05. All statistical analyses were performed using the GraphPad Prism^®^ version 5.00 program for Windows (GraphPad Software, San Diego, CA, USA, www.graphpad.com accessed on 9 February 2024).

## 3. Results

### 3.1. Immunohistochemical Tests against NFκB

The study of the intracellular localization of NFκB was performed using immunohistochemistry in the *substantia nigra* (Figure 1), jejunum (Figure 2), and colon (Figure 3). Neurons in the substantia nigra, rat jejunum, and colon enterocytes showed immunoreactivity for NFκB that increased their nuclear localization in the three structures studied from 7 days to 90 days of exposure to O_3_.

### 3.2. Lipid Oxidation: 4-Hydroxynonenal

Lipid peroxidation indices were assessed by Western blot with an antibody against 4HNE. Using the Mann–Whitney U test, the results for the substantia nigra showed a significant increase at 7, 15, and 90 days of exposure to O_3_ with respect to the control group (Figure 4A). The jejunum presented a significant increase in 4HNE at 15, 30, 60, and 90 days of exposure compared with the control group (Figure 4B). The colon presented a significant increase in 4HNE at 7, 60, and 90 days when comparing the groups treated with O_3_ with the control group (Figure 4C). All significant differences were considered with a *p* ≤ 0.05.

### 3.3. Determination of Cu/Zn-SOD in Substantia Nigra, Jejunum, and Colon

The relative abundance of Cu/Zn-SOD proteins was evaluated by Western blot (Figure 5). The densitometric analysis showed a statistically significant increase with the Mann–Whitney U test (*p* ≤ 0.05) in Cu/Zn-SOD levels at 30, 60, and 90 days of exposure to O_3_ in the substantia nigra and jejunum compared with the control group (Figure 5A,B). The results obtained for the colon showed a significant increase (*p* ≤ 0.05) in the protein levels after 30 days of treatment compared with the control group (Figure 5C).

### 3.4. Determination of Mn-SOD in Substantia Nigra, Jejunum, and Colon

Using the Mann–Whitney U test, a densitometric analysis of the Mn-SOD protein showed a significant decrease (*p* ≤ 0.05) at 7 and 60 days of exposure in the substantia nigra (Figure 6A). However, a significant decrease can be seen at 90 days for the jejunum and colon (*p* ≤ 0.05) (Figure 6B,C).

### 3.5. SOD Activity in Substantia Nigra, Jejunum, and Colon

The total activity of SOD in the substantia nigra of the rat shows us that these enzymes tend to increase their activity after 7 days of exposure to O_3_ and significantly decrease their activity at 30 days compared with control animals (Figure 7). The SOD activity in the jejunum shows that the observed changes are not statistically significant. However, the activity of total SOD in the colon decreases at 7, 15, 60, and 90 days of exposure to O_3_ compared with the control group (*p* ≤ 0.05).

## 4. Discussion

Numerous studies have shown that inhaling low doses of O_3_ repeatedly can lead to a chronic state of oxidative stress. This is because antioxidant systems are unable to counteract low and repeated doses of ROS that result from exposure to this gas [2,28]. However, the opposite occurs in the respiratory burst, in which ROS are signaling agents that activate the defense response of the immune system to fight pathogens. In this case, the antioxidant systems play a crucial role in restoring the redox balance and contributing to limiting the inflammatory response [31,32,33]. By contrast, the chronic state of oxidative stress and the loss of regulation in the inflammatory response are critical factors in maintaining a progressively degenerative process, as occurs in many chronic diseases [4,8,28]. Our results demonstrate that the loss of regulation in the inflammatory response impacts the oxidative damage found in the intestine and *substantia nigra* since they have a close relationship in neurodegenerative diseases such as Parkinson’s disease [34]. These results indicate that there is a complex relationship between both regions since together they show changes in the translocation of NFκB (Figure 1, Figure 2 and Figure 3) and alterations in 4HNE (Figure 4), SOD Cu/Zn proteins (Figure 5), SOD-Mn (Figure 6), and SOD activity (Figure 7).

The results of this work show that repeated exposure to low doses of O_3_ increases the translocation of NFκB to the cell nucleus as O_3_ treatment takes place (Figure 1, Figure 2 and Figure 3), both in the substantia nigra and the jejunum and colon, producing an increase that can be seen in immunoreactivity to NFκB in the substantia nigra shown in Figure 1A, as well as in Figure 1(B1,B2). The decrease in NFκB in the cytoplasm and the translocation of NFκB to the nucleus can also be clearly observed in the jejunum (Figure 2(B1,B2)) and colon (Figure 3(B1,B2)). However, the NFκB transcription factor, when translocated to the cell nucleus, initiates the synthesis of different cytokines and plays a crucial role in the inflammatory response [17,35]. Furthermore, 4HNE in the *substantia nigra* reached its maximum at 90 days of exposure to O_3_ (Figure 4A). Therefore, this indicates a significant increase in peroxidized lipids in this brain structure as the time of exposure to O_3_ increases. A similar effect is presented in the jejunum and colon. While in the jejunum, there is a significant increase from 15 to 90 days (Figure 4B), in the colon, this increase occurs at 7, 60, and 90 days of exposure (Figure 4C); this suggests that the higher the exposure to O_3_, the more significant the increase in peroxidation lipids and ROS levels. The rise in 4HNE alters its permeability, leading to an increase in the release of proinflammatory cytokines [36] and probable alterations in the microbiota [37], which is affected as the treatment progresses. In certain models, it was found that the expression of tight junction cellular proteins in the mouse colon is inhibited by 4HNE, which generates the translocation of NFκB to the nucleus through TLR4 receptors for proinflammatory cytokine signaling [38,39]. Our results demonstrate that the loss of regulation in the inflammatory response impacts oxidative damage found in the intestine and *substantia nigra* since they have a close relationship in neurodegenerative diseases such as Parkinson’s disease [24,34].

Work carried out at our laboratory using this same model has indicated that an irreversible neurodegenerative process is triggered after 30 days of exposure to O_3_ [40]. One of the endogenous antioxidant systems that plays an important role in counteracting ROS is SOD. The results obtained with Cu/Zn-SOD show a significant increase in this enzyme after 30 days of exposure to O_3_ (Figure 5A–C). However, the results of the total activity of SOD show that its activity significantly decreases compared with controls (Figure 7A–C). Furthermore, when we observe the effect of exposure to O_3_ on Mn-SOD, this mitochondrial enzyme also shows decreased levels, as shown in Figure 6A–C, which may partly explain the inability of this enzyme to contend with ROS under these conditions. There is a close interrelationship between SOD and NFκB that regulates the expression of a large number of genes [20,23,41,42]. Therefore, we can infer that the increase in the chronic state of oxidative stress induces an increase in cytokines by increasing the translocation of NFκB to the nucleus (Figure 1, Figure 2 and Figure 3). Oxidative stress-induced NFκB activation triggers proinflammatory gene expression [43,44]. Thus, this leads to the increased production of inflammatory mediators such as cytokines (e.g., TNF-α and IL-1β), chemokines, and adhesion molecules [45].

Furthermore, oxidative stress can modulate NFκB activity by influencing the cell’s redox state. ROS can directly modify cysteine residues within NFκB, affecting its DNA-binding ability and transcriptional activity. Oxidative modifications can also occur in signaling components upstream of the NFκB pathway, leading to altered NFκB activation dynamics and target gene expression [17].

As mentioned above, during redox balance, the formation of ROS, as a product of metabolism and the defense reactions of the immune system, is a signal for different cellular functions. However, in the presence of a chronic state of oxidative stress, redox signaling changes due to an increase in reactive species, and this leads to unregulated responses from different systems, especially a loss of control in the immune system, which generates an inflammatory reaction that has lost its regulation [2,6,13].

The results obtained show that exposure to O_3_ not only causes damage to brain structures such as the *substantia nigra* but the jejunum and colon are also sensitive to damage caused by ROS. In addition to everything that has been described regarding the relationship between the intestine, microbiota, and degenerative diseases, this has implications for the levels of regulation of other signals between the digestive system and food intake, inflammatory bowel diseases, malabsorption syndromes, etc. [12,46,47]. However, in chronic diseases, some questions still need to be resolved, and they are the following: Is O_3_ pollution capable of causing changes in the intestinal wall due to a chronic process of oxidative stress? The results obtained demonstrate that exposure to O_3_ causes an inflammatory response in the intestine that also affects the *substantia nigra*. Is O_3_ pollution capable of generating simultaneous alterations in the microbiota and the intestinal wall, which is implicated in degenerative diseases? To resolve this question, we need to perform more experiments using this model to analyze the microbiome. However, reports show that the microbiota is associated with oxidative stress in inflammatory bowel diseases [37]. Considering the results of this work, repeated exposure to low doses of O_3_ could be a critical factor in triggering other pathological conditions such as obesity, diabetes, degenerative diseases, autoimmune diseases, etc., since chronic oxidative stress would be changing pathways.

The relationship between SOD and NFκB seems to be a key player in modulating the inflammatory response in order to contend with the reactive species secondary to O_3_ exposure. The activity of antioxidant systems seems dependent on avoiding the loss of the oxidation–reduction balance. During this balance, redox signaling maintains homeostasis and the correct regulation of physiology at the molecular, biochemical, cellular, organ, and system levels. However, small and chronic changes, such as those that occur with environmental contamination caused by O_3_, gradually lead the system to an irreversible loss of redox balance and a loss of regulation in the inflammatory response, forming a vicious circle between oxidation and inflammation that accompanies the deterioration of chronic degenerative diseases [6]. O_3_ pollution causes oxidative stress, which can affect the antioxidant effects of SOD by inhibiting its activity and ultimately reducing its expression, thereby compromising the cell’s antioxidant defense system. Furthermore, ozone pollution can also activate NFκB and modulate its translocation to the nucleus, leading to inflammatory altered responses. Therefore, the inflammatory alterations found in the jejunum, colon, and substantia nigra are a direct result of repeated exposure to low doses of ozone.

## 5. Conclusions

With the results obtained in this work, we can conclude the following: Oxidative stress caused by low ozone exposure doses in rats generates alterations in both the intestinal wall and the *substantia nigra*. These effects are characterized by a loss of regulation in the inflammatory response, as shown by the alterations in NFkB, as well as in endogenous antioxidant systems such as SOD, both in the intestine and the *substantia nigra*. Therefore, the close relationship between the intestine and the brain through the nervous system, mainly through the *vagus* nerve, hormones, and cytokines, leads to oxidative alterations caused by environmental ozone pollution in both systems, closely related to degenerative processes.

## Figures and Tables

**Figure 1 antioxidants-13-00536-f001:**
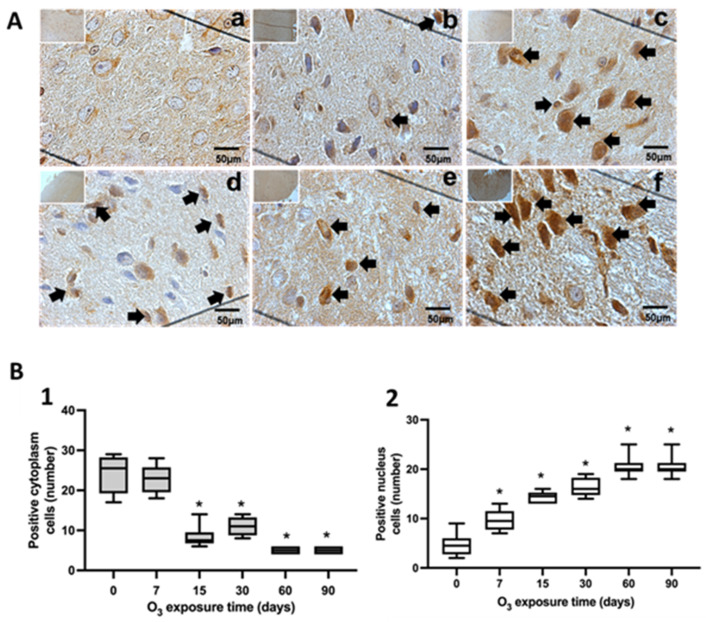
Effect of chronic exposure to low doses of O_3_ on rat substantia nigra neurons. (**A**) Micrographs of NFκB in substantia nigra of rats exposed to O_3_: (**a**) control, (**b**) 7 days of O_3_ exposure, (**c**) 15 days of O_3_ exposure, (**d**) 30 days of O_3_ exposure, (**e**) 60 days of exposure to O_3_, and (**f**) 90 days of exposure to O_3_. The micrograph shows an increase in NFκB immunoreactivity in the nucleus at 15, 30, 60, and 90 days of exposure to O_3_. Arrows indicate the localization of NFκB in the cell nucleus (calibration bar = 50 µm; 100× photomicrograph). (**B**) The abscissa axis shows the different O_3_ exposure treatments for the animals, and the ordinate axis shows the median number of cells reactive to NFκB. * *p* ≤ 0.05. (**1**) The graph shows the effect of chronic exposure to low doses of O_3_ on the localization of NFκB in the cell cytoplasm. The results show a cell decrease from 15 to 90 days of exposure to O_3_ (*p* ≤ 0.05). (**2**) The graph shows the effect of chronic exposure to low doses of O_3_ on the localization of NFκB in the cell nucleus, noting the increase in the nuclear localization of NFκB.

**Figure 2 antioxidants-13-00536-f002:**
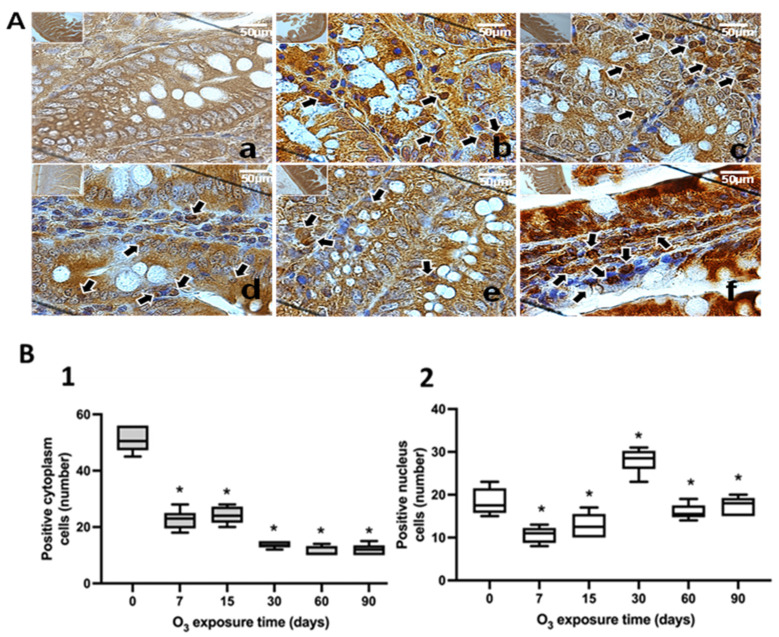
Effect of chronic exposure to low doses of O_3_ on enterocytes in the jejunum of rats. (**A**) Micrographs of NFκB in the jejunum of rats exposed to O_3_: (**a**) control, (**b**) 7 days of O_3_ exposure, (**c**) 15 days of O_3_ exposure, (**d**) 30 days of O_3_ exposure, (**e**) 60 days of exposure to O_3_, and (**f**) 90 days of exposure to O_3_. The micrograph shows an increase in NFκB immunoreactivity in the nucleus at 7, 15, 30, 60, and 90 days of exposure to O_3_. Arrows indicate the localization of NFκB in the cell nucleus (calibration bar = 50 µm; 100× photomicrograph). (**B**) The graph shows the different exposure treatments to O_3_ for the animals on the abscissa axis and the median number of cells reactive to NFκB on the ordinate axis. * *p* ≤ 0.05. (**1**) The graph shows the effect of chronic exposure to low doses of O_3_ on the localization of NFκB in the cell cytoplasm; a decrease in the number of cells is observed from 7 to 90 days of exposure to O_3_ (*p* ≤ 0.05). (**2**) The graph shows the number of cells with the NFκB label in the cell nucleus; the results show an increase in the number of cells from 7 to 90 days of exposure to O_3_.

**Figure 3 antioxidants-13-00536-f003:**
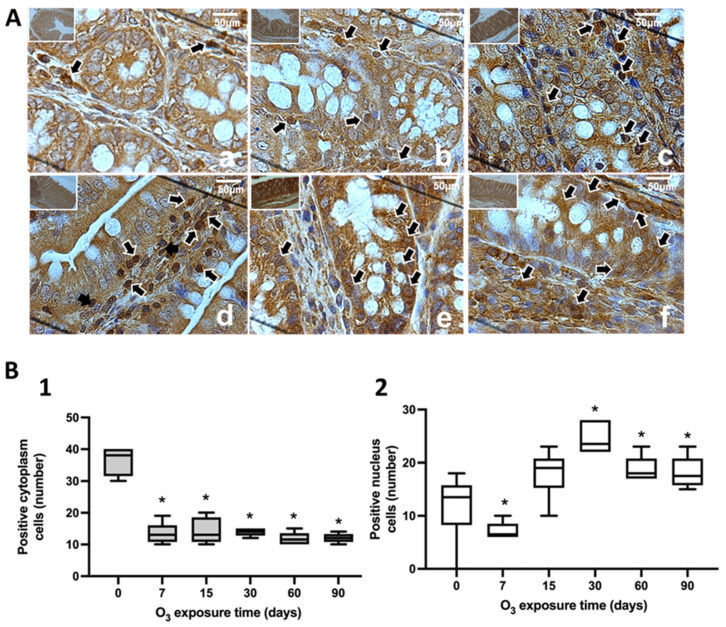
Effect of chronic exposure to low doses of O_3_ on enterocytes in the colons of rats. (**A**) Micrographs of NFκB in the colons of rats exposed to O_3_: (**a**) control, (**b**) 7 days of O_3_ exposure, (**c**) 15 days of O_3_ exposure, (**d**) 30 days of O_3_ exposure, (**e**) 60 days of exposure to O_3_, and (**f**) 90 days of exposure to O_3_. The micrograph shows an increase in NFκB immunoreactivity in the nucleus at 15, 30, 60, and 90 days of exposure to O_3_. Arrows indicate the localization of NFκB in the cell nucleus (calibration bar = 50 µm; 100× photomicrograph). (**B**) The graph shows the different animal exposure treatments to O_3_ regarding the abscissa axis and the median number of cells reactive to NFκB on the ordinate axis. * *p* ≤ 0.05. (**1**) The graph shows the effect of chronic exposure to low doses of O_3_ on the localization of NFκB in the cell cytoplasm; note a decrease in the number of cells from 7 to 90 days of exposure to O_3_ (*p* ≤ 0.05). (**2**) The graph shows the effect of chronic exposure to low doses of O_3_ on nuclear localization; the results show a significant increase in the number of cells from 7 to 90 days of exposure to O_3_.

**Figure 4 antioxidants-13-00536-f004:**
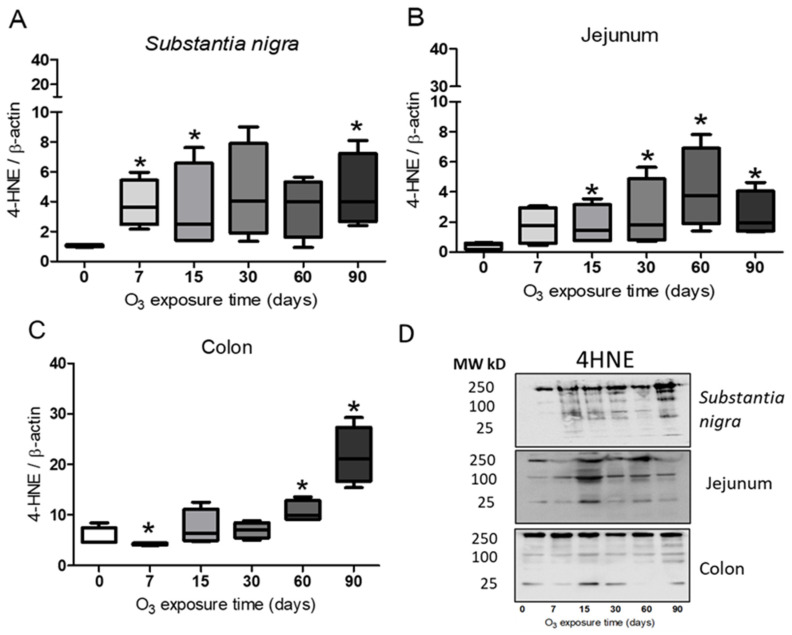
Effect of exposure to O_3_ on lipid oxidation. The abscissa axis shows the treatment, and the ordinate axis shows the relative abundance of 4HNE represented in arbitrary units. * *p* ≤ 0.05. (**A**) Substantia nigra. The results show a significant increase in 4HNE at 90 days compared with the control group. (**B**), Jejunum. The graph shows a significant increase in 4HNE after 30, 60, and 90 days compared with the control group. (**C**) Colon presented an increase in 4HNE at 60 and 90 days compared with the control group. (**D**) Representative images for band analysis.

**Figure 5 antioxidants-13-00536-f005:**
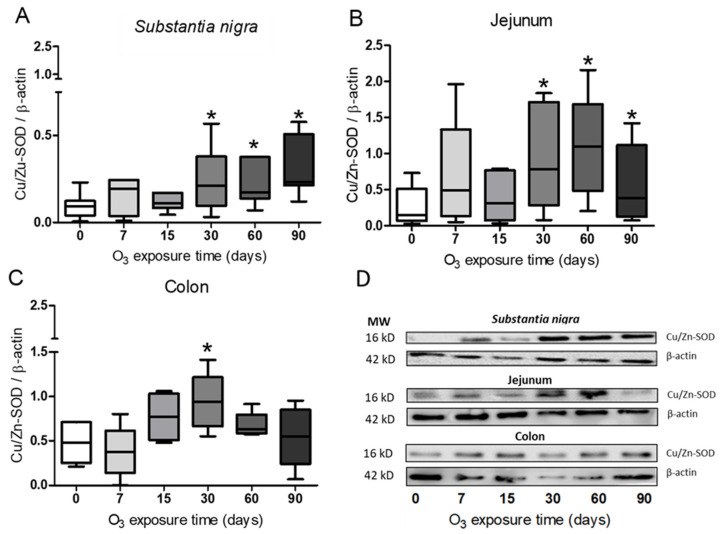
Effect of exposure to O_3_ on the relative abundance of Cu/Zn-SOD. The abscissa axis shows the O_3_ treatments, and the ordinate axis shows the relative abundance of Cu/Zn-SOD represented in arbitrary units. * *p* ≤ 0.05. (**A**) In the *substantia nigra* and (**B**) the jejunum, a significant increase can be observed at 30, 60, and 90 days of exposure to O_3_ compared with the control group, (**C**) while for the colon, an increase is shown at 30 days compared with the control group. (**D**) Representative WBs for each treatment.

**Figure 6 antioxidants-13-00536-f006:**
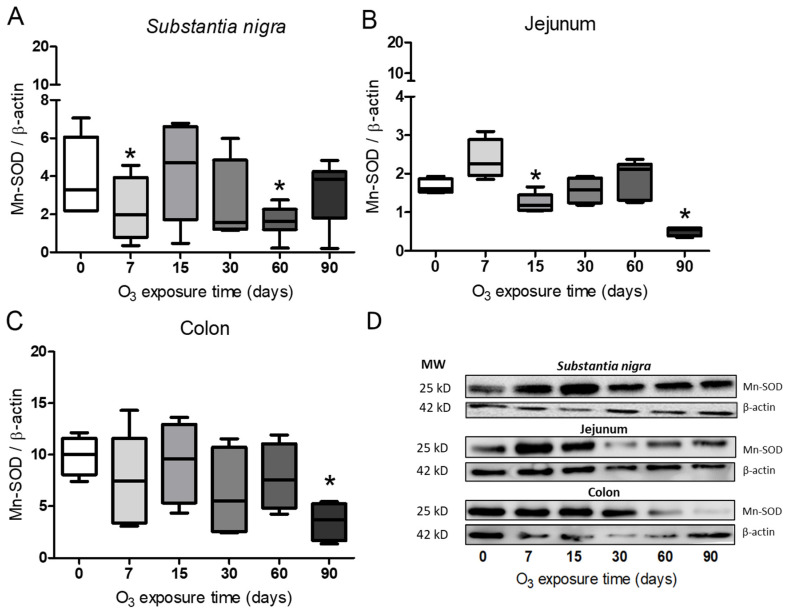
Effect of exposure to O_3_ on the relative abundance of Mn-SOD. The abscissa axis shows the O_3_ treatments, and the ordinate axis shows the relative abundance of Mn-SOD represented in arbitrary units. * *p* ≤ 0.05. (**A**) Substantia nigra. The Mn-SOD protein decreased at 7 and 60 days and at 90 days in the jejunum and colon, respectively (**B**,**C**). (**D**) Representative WB for each condition.

**Figure 7 antioxidants-13-00536-f007:**
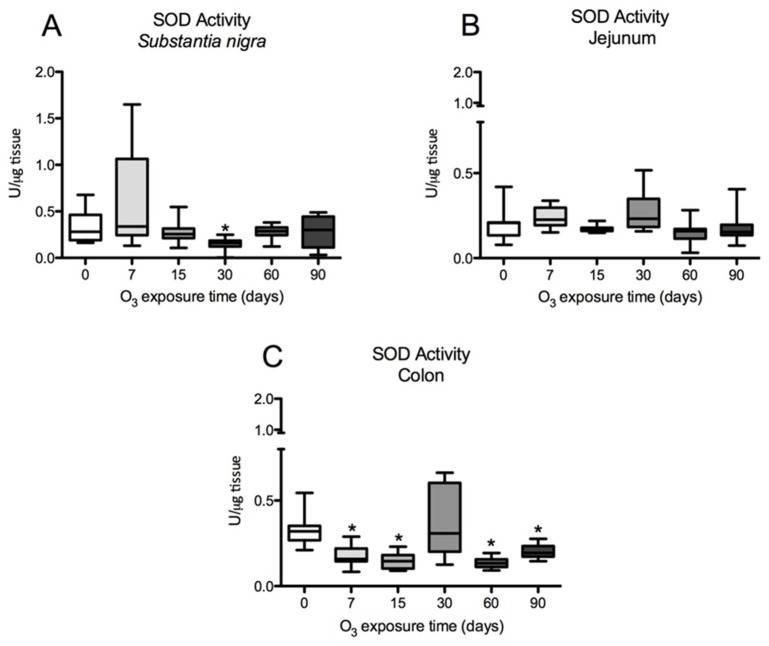
Effect of exposure to O_3_ on SOD activity in the *substantia nigra*, jejunums, and colons of rats (* *p* ≤ 0.05). The abscissa axis shows the different O_3_ exposure treatments used on the animals, and the ordinate axis shows the SOD units per microgram of tissue. (**A**) Total SOD enzyme activity in the substantia nigra shows a significant decrease at 30 days of O_3_ exposure compared with the control group. (**B**) SOD activity in the jejunum does not show statistically significant variations. (**C**) In the rat colon, total SOD activity is significantly decreased at 7, 15, 60, and 90 days of O_3_ exposure compared with control animals. A Mann–Whitney U test was used to compare the groups against their respective controls.

## Data Availability

The data in this work will be available when required.

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
