# Peer review of "Changes in SOD and NF-κB Levels in Substantia Nigra and the Intestine through Oxidative Stress Effects in a Wistar Rat Model of Ozone Pollution"

_antioxidants, 2024, doi:10.3390/antiox13050536_

Round 1

Reviewer 1 Report

Dear Authors, dear Editor,

Draft “antioxidants-2948414-v1 Changes on SOD and NF-kappaB levels“ reports on a rat experiment of inhalation of Ozone (0.25 ppm of O3 for 4 h daily, put the O3 concentration in the abstract too) and for variable length of time (0 90 days). Examined biomarkers measured in tissue homogenates of substantia nigra, jejunum and colon include the HNE lipid oxidation product, Cu/Zn and Mn SOD enzymes, protein receptor and transcription factor NfkB. Results obtained by the Authors show that “exposure to O3 not only causes damage to brain structures such as the substantia nigra, but the jejunum and colon are also sensitive to damage caused by ROS”. Previous work of the Authors with the same murine model highlighted irreversible neurodegenerative process is triggered after 30 days of exposure to O3 (where is it cited? L 309 Put it here). This research report is complete and neatly drafted, therefore it is well worth publishing in Antioxidants. BTW, it is a rat, or rodent experiment in the title, not “murine”, which is for the mouse. In the Materials and Methods section, just for logic, put Immunohistochemistry just after exposure: you have to put down the rats before doing any analysis.

However, some claims in the discussion, which make the core of the Authors’ draft, are at this point undemonstrated and speculative: “Is O3 pollu-351 tion capable of causing changes in the intestinal wall due to a chronic process of oxidative 352 stress? The results obtained demonstrate that exposure to O3 causes an inflammatory re-353 sponse in the intestine which also affects the substantia nigra. Is O3 pollution capable of 354 generating simultaneous alterations in the microbiota and the intestinal wall, implicated 355 in degenerative diseases?”.

I am in charge of updating a twice-yearly toxicology report on Ozone and other environmental and workplace pollutants. How should I discuss this particular work, once it is published? Apart from SODs in tissues, there is no other biomarker measurement of oxidative stress, while several would be available for measuring in 72 rats. Were they measured and reported elsewhere? If it is so, I couldn’t find the citation. Otherwise, I find some degree of mis-planning in the choice of indicators. Given that exposure is from inhalation, I would expect circulatory and lung tissue biomarkers being included, as reference for those measured in the other, rather exotic, tissues. This makes a harder point in my review, since the experimental plan is somewhat flawed from the start, unless this is only a part of a wider project, where more biomarkers and tissues, blood included, were tested: in this, more favourable case, you should have it stated. Of course, I can’t recommend rejection on the ground of this experimental bias; however, I would point this to you as experimentalists, as a comment. If you cannot complete the job, I have little other objections to publication after minor adjustments.

Kind regards

See major comments

Author Response

Dear reviewer,

We are immensely grateful for the corrections you provided for our work. Your input has been incredibly valuable in helping us to correct our mistakes, clarify our article, and enhance our methodology and discussion. As authors, we would like to express our sincere appreciation for the time and effort you have devoted to improving our manuscript.

Thank you very much.

Sincerely

Dr. Selva Rivas-Arancibia

1.- Question: “Exposure to O3 not only causes damage to brain structures such as the substantia nigra, but the jejunum and colon are also sensitive to damage caused by ROS”. Previous work of the authors with the same murine model highlighted irreversible neurodegenerative process is triggered after 30 days of exposure to O3 (where is it cited? Put it here).

Answer

Exposure to ozone affects the whole organism, but it mainly affects the organs and brain structures most sensitive to oxidative stress. We have previously established the appropriate dose and duration of ozone exposure. Additionally, we have discovered that the damage becomes permanent after 30 days of exposure and cannot be reversed. At the 2002 Society for Neurosciences Congress, it was revealed that animals experience increasing deterioration after 30 days, even without exposure to ozone. This information was placed in the indicated line (line XXX).

2- Question: “It is a rat or rodent experiment in the title, not “murine,” which is for the mouse.”

Answer: Both rats and mice are classified under the murine family by phylogenetic classification. Therefore, some authors refer to both as murine. In our study, we have specifically utilized Wistar rats from the Murine family. Hence, we have used a murine experimental model. We will be sending you additional information about our study. However, we are willing to change the title of our research to reflect our focus accurately.

New title: Changes on SOD and NF-kappaB levels in substantia nigra and intestine, by oxidative stress effects in a Wistar rat model of ozone pollution

Phylum: Chordates

Class: Mammals

Order: Rodentia

Suborder: Myomorpha

Family: Muridae

Subfamily: Murinae

Genus: Rattus

Species: Rattus norvergicus

Strain: Wistar

Taken from: Modlinska K, Pisula W. The Norway rat, from an obnoxious pest to a laboratory pet. Elife. 2020 Jan 17;9:e50651. doi: 10.7554/eLife.50651. PMID: 31948542; PMCID: PMC6968928.

3.- Question: “In the Materials and Methods section, just for logic, put Immunohistochemistry just after exposure: you have to put down the rats before

doing any analysis.”

Answer: Thank you very much, the correction was done.

4.- Question: “Is O3 pollution capable of causing changes in the intestinal wall due to a chronic process of oxidative stress?

Answer: Repeated exposure to low doses of ozone causes a state of chronic oxidative stress. Studies show that this redox balance alteration can alter the inflammatory response regulation, causing endothelial cell inflammation and potentially leading to loss of function.

One of the clearest demonstrations mentioned in the introduction is that gastrointestinal pathological processes are characterized by inflammation and loss of intestinal wall permeability.

5.- Question: Is O3 pollution capable of generating simultaneous alterations in the microbiota and the intestinal wall, implicated in degenerative diseases?”.

Answer: You are most likely right. We are studying temporal changes in the microbiota of rats exposed to low ozone doses. We have the samples; we need to run the experiments.

However, we do not know the temporal relationship between intestinal wall and microbiota changes. I think this model has changes in the microbiota, but that would be speculative, so we will wait for the results.

6.- Question: How should I discuss this particular work once it is published?

Answer: This study presents compelling evidence of the link between the gut and the brain. It suggests that the gut can be affected by repeated exposure to environmental ozone pollution. It also demonstrates the relationship between the loss of the activity of a critical antioxidant enzyme, such as superoxide dismutase, and the inflammatory response. It is evident that the response we are observing does not have any self-limiting mechanism. This fact is because one of the critical elements that triggers the production of cytokines is the translocation factor NFkB, which continues to increase its presence in the nucleus throughout the experiment. In addition, it indicates a persistent increase in cytokine synthesis, and any self-regulation is not present.

Therefore, repeated exposure to low ozone doses leads to impaired regulation of the inflammatory response and the subsequent damage it can cause.

7.- Question: Apart from SODs in tissues, there is no other biomarker measurement of

oxidative stress, while several would be available for measuring in 72 rats. Were they

measured and reported elsewhere? If it is so, I couldn’t find the citation.

Answer: We have been working on a broader research line that involves the effect of ozone pollution on various aspects related to oxidative stress markers, like peroxidized lipids and oxidized proteins. We have also analyzed changes in antioxidant systems such as glutathione. Our research focused on cytokines alteration, including Th-1, Th-2, and Th-17 responses in the brain structures related to neurodegenerative diseases. We also analyzed some interleukins present in the blood. Our main works were published in international journals.

We have submitted a paper to the International Journal of Molecular Sciences on the involvement of alpha-synuclein in the inflammatory response in the intestine and brain.

8. Question: I would expect circulatory and lung tissue biomarkers to be included as references for those measured in the other, rather exotic, tissues.

Answer: Many works by other authors have measured these markers in the lungs and blood. We will include them in the discussion of this paper.

9. Question: This is a harder point in my review since the experimental plan is somewhat flawed from the start unless this is only part of a wider project, where more biomarkers and tissues, including blood, were tested: in this case, it is a more favourable case.

Answer: As we previously explained, this is a part of extensive research that has been studied for a long time. Our research focuses on the modifications of Th-1, Th-2, and Th-17 responses in various brain structures related to neurodegenerative diseases. Additionally, we have analyzed some interleukins present in the blood, and all our findings have been published in international journals.

Reviewer 2 Report

This study shows that a lipid oxidation and the changes in superoxide dismutases could be caused by exposure to the ozone in brain and intestine. However, the reviewer thought that antioxidants which could counteract those degeneration should be explored.

Other comments are;

1. Inflammatory factors, such as cytokines, should be examined, too, in order to elucidate tissue damages.

2. The “conclusions” is a very speculation. More evidences of tissue degeneration should be shown in this study.

3. Explain the generation method of ozone in more detail, please.

4. Are mice at 0 day as old as mice at 90 days in each experiment? The reviewer concerned whether mice of same week of age were used.

This study shows that a lipid oxidation and the changes in superoxide dismutases could be caused by exposure to the ozone in brain and intestine. However, the reviewer thought that antioxidants which could counteract those degeneration should be explored.

Other comments are;

1. Inflammatory factors, such as cytokines, should be examined, too, in order to elucidate tissue damages.

2. The “conclusions” is a very speculation. More evidences of tissue degeneration should be shown in this study.

3. Explain the generation method of ozone in more detail, please.

4. Are mice at 0 day as old as mice at 90 days in each experiment? The reviewer concerned whether mice of same week of age were used.

Author Response

Dear reviewer,

We greatly appreciate the time dedicated to correcting this article. Your work helped us correct errors, clarify the manuscript, and justify our discussion better.

Thank you very much for your generosity.

Sincerely

Dr. Selva Rivas-Arancibia

This study shows that a lipid oxidation and the changes in superoxide dismutases could be caused by exposure to the ozone in brain and intestine. However, the reviewer thought that antioxidants which could counteract those degeneration should be explored.

Other comments are;

  1. Inflammatory factors, such as cytokines, should be examined, too, in order to elucidate tissue damages.

Answer: In this publication they did not address these cytokines, since it is part of a broader line of research and these results of interleukins in the intestine will be included in an exclusive manuscript to evaluate their inflammatory profile. However, in Nervous System we already have published information on the matter. We also have a publication under review where the role of IL17 in the brain and intestine is observed.

Our research focused on cytokines alteration, including Th-1, Th-2, and Th-17 responses in the brain structures related to neurodegenerative diseases. We also analyzed some interleukins present in the blood. Our main works were published in international journals.

Solleiro-Villavicencio, H.; Hernández-Orozco, E.;Rivas-Arancibia, S. Effect of exposure to low doses of ozone on interleukin 17A expression during progressive neurodegeneration in the rat hippocampus. Neurologia (Engl Ed). 2021, 36(9):673-680. doi:10.1016/j.nrleng.2018.08.003.

Velázquez-Pérez, R.;Rodríguez-Martínez, E.;Valdés-Fuentes, M.;Gelista-Herrera, N.;Gómez-Crisóstomo, N.;Rivas-Arancibia, S. Oxidative Stress Caused by Ozone Exposure Induces Changes in P2X7 Receptors, Neuroinflammation, and Neurodegeneration in the Rat Hippocampus. Oxid Med Cell Longev. 2021, 20213790477. doi:10.1155/2021/3790477.

Marlen Valdés-Fuentes, Erika Rodríguez-Martínez  and Selva Rivas-Arancibia. Accumulation of Alpha-synuclein and increase in the inflammatory response in the substantia nigra, jejunum, and colon, in a model of O3 pollution in rats. Send to International Journal of Molecular Sciences.

  1. The “conclusions” is a very speculation. More evidences of tissue degeneration should be shown in this study.

Answer: To avoid speculation, we have included the required citations in the discussion section of this article.

  1. Explain the generation method of ozone in more detail, please.

Answer: Animals were enclosed daily for 4 h in a chamber with a diffuser connected to a variable flux ozone generator (5 l/s). Ozone was generated from a tube through which a high-voltage current circulated. The tube contained aluminum chips with two electrodes inside that allowed the conversion of oxygen that circulated around the tube into ozone. The air, feeding the ozone converter, was filtered purified air. Ozone production levels were proportional to the current intensity and airflow. A PCI Ozone & Control System Monitor approved by the Environmental Protection Agency was used to measure the ozone concentration inside the chamber throughout the experiment and ozone concentration was kept constant.

Air exposure

The same chamber was used when treating the control group where flow of ozone-free purified air was used.

This study is part of a long-term research project. The first article on this topic was published in 1998. Since then, we have developed the model we are currently studying, published in 2006.

Pereyra-Muñoz, N.;Rugerio-Vargas, C.;Angoa-Pérez, M.;Borgonio-Pérez, G.;Rivas-Arancibia, S. Oxidative damage in substantia nigra and striatum of rats chronically exposed to ozone. J Chem Neuroanat. 2006, 31(2):114-123. doi:10.1016/j.jchemneu.2005.09.006.

  1. Are mice at 0 day as old as mice at 90 days in each experiment? The reviewer concerned whether mice of same week of age were used.

Answer:  We use the Wistar rat, which is part of the Murinae family. To avoid confusion, we modify the title to read:

“Changes on SOD and NF-kappaB levels in substantia nigra and intestine, by oxidative stress effects in a Wistar Rat model of ozone pollution”

In our study on the effects of repeated exposure to low doses of ozone, each group of rats was exposed to air during the exact times of exposure to ozone, and a separate control group was maintained for each experimental group. We found no significant differences between the control groups. To establish a baseline for comparison, we utilized groups that were subjected to air exposure for 30 days as our control group. Thank you very much in the method was clarified.

Round 2

Reviewer 2 Report

The reviewer appreciates any response and effort which has been done by the authors to enhance the quality of the work. After the check and correction of misspelling is carefully done again, the manuscript would be acceptable.

The reviewer appreciates any response and effort which has been done by the authors to enhance the quality of the work. After the check and correction of misspelling is carefully done again, the manuscript would be acceptable.